# Bcl-xL: A Focus on Melanoma Pathobiology

**DOI:** 10.3390/ijms22052777

**Published:** 2021-03-09

**Authors:** Anna Maria Lucianò, Ana B. Pérez-Oliva, Victoriano Mulero, Donatella Del Bufalo

**Affiliations:** 1Departamento de Biología Celular e Histología, Facultad de Biología, Universidad de Murcia, IMIB-Arrixaca, Centro de Investigación Biomédica en Red de Enfermedades Raras, 30100 Murcia, Spain; annamaria.luciano@um.es; 2Preclinical Models and New Therapeutic Agents Unit, Department of Research and Advanced Technologies, IRCCS Regina Elena National Cancer Institute, 00144 Rome, Italy

**Keywords:** Bcl-xL, melanoma, cancer, invasion, migration, apoptosis

## Abstract

Apoptosis is the main mechanism by which multicellular organisms eliminate damaged or unwanted cells. To regulate this process, a balance between pro-survival and pro-apoptotic proteins is necessary in order to avoid impaired apoptosis, which is the cause of several pathologies, including cancer. Among the anti-apoptotic proteins, Bcl-xL exhibits a high conformational flexibility, whose regulation is strictly controlled by alternative splicing and post-transcriptional regulation mediated by transcription factors or microRNAs. It shows relevant functions in different forms of cancer, including melanoma. In melanoma, Bcl-xL contributes to both canonical roles, such as pro-survival, protection from apoptosis and induction of drug resistance, and non-canonical functions, including promotion of cell migration and invasion, and angiogenesis. Growing evidence indicates that Bcl-xL inhibition can be helpful for cancer patients, but at present, effective and safe therapies targeting Bcl-xL are lacking due to toxicity to platelets. In this review, we summarized findings describing the mechanisms of Bcl-xL regulation, and the role that Bcl-xL plays in melanoma pathobiology and response to therapy. From these findings, it emerged that even if Bcl-xL plays a crucial role in melanoma pathobiology, we need further studies aimed at evaluating the involvement of Bcl-xL and other members of the Bcl-2 family in the progression of melanoma and at identifying new non-toxic Bcl-xL inhibitors.

## 1. Introduction

Throughout their evolution, multicellular organisms have developed the ability to eliminate damaged or unwanted cells with the aim to maintain a continuous homeostasis of tissues. This constant turnover is essential to the development of healthy tissues and organisms. Cells are eliminated through regulated cell death mechanisms including apoptosis, necrosis, and autophagy [1]. These mechanisms have been developed in all the vertebrates and have preserved well with evolution, securing an orderly development of organisms [2,3] whilst preventing the onset of diseases, including cancer [4,5]. Specifically, impaired apoptosis represents a hallmark of cancer [6].

The apoptotic pathways have at the basis of the process the activation of caspases, that degrade cellular components in order to be recognized by phagocytes [7,8]. Caspases are part of a family of cysteine proteases preserved throughout evolution and involved in cell death, as well as in the inflammatory response [9,10]. The apoptotic caspases can be divided into “initiator” (caspases 8, 9, 10) and “effector” (caspases 3, 6, 7) caspases [10]. Under physiological conditions, caspases are present in the form of inactive pro-caspases, which are activated in a cascade process that ends up with the activation of the effector caspases through proteolytic cleavage and consequent induction of cell death [11,12,13]. For such process to take place, the cell must receive a series of external or internal stimuli that activate the apoptotic pathway. Two different pathways in the activation of this process are known to date, thereof one is commonly defined as “intrinsic” and mediated by mitochondria, and the other one is known as “extrinsic” and mediated by death receptors [14]. The extrinsic pathway is induced by death ligands produced by patrolling cells, and for this reason, it is also known as the death receptor pathway of apoptosis [15]. After death ligand binding to the death receptors expressed in the target cell membrane, the extrinsic pathway is induced through the activation of caspase 8 [1]. In contrast, the intrinsic process can be induced by all cells in response to various stress stimuli: examples include oxidative and endoplasmic reticulum stress, nutrient deprivation, pathogen infection or DNA damage [16]. The key regulators of this process are members of the B-cell lymphoma 2 (Bcl-2) family which, following activation, cause at first a permeabilization of the mitochondrial outer membrane, and then a release of cytochrome C from the mitochondria into cytosol, assembly of the apoptotic protease activating factor-1 (APAF-1) forming apoptosome by oligomerization and at last, activation of the caspase cascade through caspases 9 [16,17].

A strong correlation between incorrect apoptotic process and various pathologies has been identified [18]. Tumor cells, for example, increase their pro-survival activity during the tumorigenic process, thus developing resistance to apoptotic stimuli [19].

## 2. Melanoma

Members of the Bcl-2 family are frequently deregulated in cutaneous melanoma, the most dangerous and deadly form of skin cancer that accounts for more than 75% of all skin cancer deaths [20,21,22,23]. For example, in 2020 in Italy, the estimated new cases of cutaneous melanoma were 100,350, representing approximately 5.6% of all cancer cases diagnosed in the same year, while the estimated deaths caused by melanoma were 6850, representing 1.1% of all cancer deaths (https://seer.cancer.gov/statfacts/html/melan.html visited on 15 February 2021).

Melanoma originates in melanocytes, the pigment-producing cells present in the basal layer of the epidermis, usually when unrepaired DNA damage triggers mutations that lead the melanocytes to multiply rapidly and form malignant tumors. Melanoma is caused mainly by intense exposure to ultraviolet radiation from sunshine or tanning beds, with acute insurgence in individuals who are genetically predisposed to the disease with the red hair color phenotype [24,25]. Other factors, such as a compromised immune system, could also cause melanoma, as well as a family history that is included among the main risk factors involved in its pathogenesis [26].

In general, melanoma has two growth stages: radial and vertical [27]. During the radial growth stage, malignant cells grow in an outward movement, spreading across the epidermis, or invading the papillary dermis. In time, most melanoma progresses to the vertical growth stage, during which the malignant cells invade the dermis and develop the ability to spread or metastasize. The majority of first metastases are local or regional sites, including regional lymph nodes. The most common distant sites where melanoma spreads include lung, liver, distant areas of the skin, brain, gastrointestinal tract, as well as bone and adrenal gland [27].

Several driver mutations have been identified in melanoma and the most common mutations affect signaling of the Ras/Raf/mitogen-activated protein kinase pathways. Also, the Bcl-2 family network is found deregulated in melanoma [28].

In early stages, melanoma can be treated with high success rates; in later stages, it can easily spread to other parts of the body, where it becomes difficult to treat, thus having a poor prognosis. Recently, different therapies have been approved by the US Food and Drug Administration (FDA) for melanoma treatment [29,30,31]. Depending on the clinical features of the tumor (e.g., location, stage, genetic profile), the therapeutic protocol may require surgical resection, chemotherapy, radiotherapy, photodynamic therapy, target therapy or immunotherapy. For patients with stage I–IIIB melanoma, surgery is the primary treatment [31]. Surgical interventions are accompanied by chemotherapy, although therapeutic treatment could induce side effects due to the toxicity in the skin or in the gastrointestinal tract, as well as immune reactions and reduced efficiency [32]. New therapeutic targets have been described from genetic studies on melanocytes and from the identification of factors involved in the pathogenesis of the malignant transformation of the melanocytic cells [33,34,35].

## 3. Bcl-2 Family

Proteins of the Bcl-2 family are essential supplements to the signals leading to cell survival or apoptosis, while the cell’s fate itself depends primarily on quantity, location and interaction among specific Bcl-2 family proteins [36].

Members of the Bcl-2 family are classified according to structure and function (Figure 1). The anti-apoptotic members of this family, which include Bcl-2, B-cell lymphoma-extra-large (Bcl-xL), B-cell lymphoma-w (Bcl-w), Bcl-2-related protein A1/Bcl-2-related isolated from fetal liver-11 (A1/Bfl-1) and myeloid cell leukemia-1 (Mcl-1) share 4 Bcl-2 homology domains (BH1-BH4). The peculiar trait shown in this group is the presence of the BH4 domain in the N-terminal [37]. Through the BH4 domain, Bcl-xL binds the pro-apoptotic proteins of the Bcl-2 family, such as BAX and BAD, thereby preventing the activation of apoptotic signaling that leads to the opening of pro-apoptotic ion channels [38]. By doing so, Bcl-xL supports cell survival by inhibiting intrinsic cell death pathway, such as the release of cytochrome C and apoptosome assembly [39,40]. In the case of Bcl-2, when a mutation on the BH4 domain occurs, this reduces the stability of the protein and interferes with the occupation of BAX [41].

As the BH4 domain is capable of binding to other proteins that do not belong to the Bcl-2 family, some anti-apoptotic members, including Bcl-xL, are capable of more than just inhibition of apoptosis to which are traditionally associated, specifically contributing to other important cellular functions, such as proliferation, autophagy, differentiation, DNA repair, tumor progression and angiogenesis [42]. Regarding their pro-survival role, anti-apoptotic proteins of the Bcl-2 family discharge their duties by binding and inhibiting pro-apoptotic proteins, cell stress sensors (proteins BH3-only) and apoptosis effectors (Bcl-2-associated X protein, BAX and Bcl-2 antagonist/killer, BAK) [36].

The pro-apoptotic proteins that belong to the Bcl-2 sub-group like BAX and BAK share the BH1-BH3 domains [43].

The third class of proteins possesses the BH3 domain: Bcl-2-interacting mediator of cell death (BIM), p53-upregulated modulator of apoptosis (PUMA) and truncated form of BH3-interacting domain death agonist (tBID) [44]. These proteins are called “activators”, as they can bind and provoke new conformations of BAX or BAK to induce MOMP, while BH3 proteins that do not associate with BAX and BAK are called “synthesizers”, like BAD or NOXA that endorse apoptosis by both directly activating BAX and BAK and by suppressing the anti-apoptotic proteins at the mitochondria and the endoplasmic reticulum [44,45,46]. The perfect functioning of these systems lies in the delicate balance between the different protein components of the Bcl-2 family: in particular, in the activity of (i) the anti-apoptotic proteins that retain the activators and the molecules that initiate MOMP, and (ii) the sensitizers, which antagonize members of the pro-survival family by releasing the only BH3-activating proteins BAX/BAK.

### 3.1. Bcl-xL: The Pro-Survival Member of the Bcl-2 Family

Bcl-xL is encoded by the *BCL2L1* gene, which is located on human chromosome 20 at band q11.21 and gives two isoforms of mature RNA through alternative splicing: the short form Bcl-xS, containing three exons, and the large form Bcl-xL, containing 4 exons [47]. At protein level, three different isoforms can be found: the longer form, containing 233 amino acid residues in length, acts as an apoptotic inhibitor; the shorter form, of 170 amino acid, acts as an activator of apoptosis; and a third form, called Bcl-xβ, containing 227 amino acids, differs from the longer and shorter forms in the last 45 amino acids, but does not display any specific functions [48] (Figure 2)**.**

Bcl-xL protein structure is made up of 8 alpha helix regions. Alpha5 and alpha6 integrate a central hairpin that is flanked by the alpha3 and alpha4 on one side, and by alpha1, alpha2 and alpha8 on the other side [49]. Looking at their three-dimensional structures, BH domains appear essential to the formation of tertiary structures. The BH1 and BH2 domains cover the regions connecting each two helices, alpha1 and alpha2 in the case of the BH1, and alpha7 and alpha8 in the case of the BH2 [49]. The BH3 domain is located entirely in the alpha2 helix, while the BH4 domain is located in the alpha1 and makes up hydrophobic contacts with alpha2, alpha5 and alpha6 [49].

A rather interesting structural element of Bcl-xL is a large hydrophobic groove involving the BH1 and BH3 domains, predominantly formed between the alpha3 and alpha4 helices (Glutamine-111 on α3 and Glutamic acid-129 on α4), and the alpha5 found at its basis. This region is deemed to be the most relevant difference among members of the pro-survival protein family, which seems to have the same conformation and little identity of sequence. In fact, other members have a v-shape structure of the helices resulting in a more open groove [50,51,52,53].

The finely regulated interaction between pro- and anti-apoptotic proteins is made possible by the spatial architecture of the BH3 domains. In fact, the hydrophobic groove appears to be responsible for the capture of the pro-apoptotic molecules. In most of the cases, the overall interaction is determined by 4 hydrophobic residues of the BH3 domain projecting into the groove and an electrostatic interaction between the conserved aspartate-83 and arginine-139 in Bcl-xL [54,55]. After interacting with the BH3 domain, the immediate reaction is an opening of the hydrophobic groove where the alpha3 shifts away and the alpha4 moves to create an opening of the groove that acquires a V-shaped form. This phenomenon is typical of Bcl-xL protein and sets it apart from other family members [49].

Through its interaction with the DNA-binding region of p53 protein, Bcl-xL also plays a relevant role in p53-induced mitochondrial permeabilization and apoptosis. Nuclear magnetic resonance spectroscopy identified the carboxy-terminus of the first alpha-helix, and the loops between alpha3/alpha4 and alpha5/alpha6 as the Bcl-xL regions involved in the binding with p53 [56,57,58].

### 3.2. Regulation of Bcl-xL

The *BCL2L1* promoter is highly conserved between human and mouse. When isolated for the first time, both murine and human promoters have shown two distinct regions with promoter activity [59]. The first promoter region is located immediately above the first codon exon, while the second one is above the first non-coding region [60]. Moreover, consensus motifs for a number of transcription factors have been identified, as Sp1, AP1, Oct-1, E26 transformation specific (ETS), Rel/Nuclear Factor-κB (NF-kB), Signal Transducer and Activator of Transcription (STATs), and GATA-1, among which STATs, Rel/NF-kB, ETS and the AP1 complex have been demonstrated to play an important role in the regulation of *BCL2L1* gene expression [47,59] (Figure 3)**.**

Appendix A shows the list of putative transcription factors able to bind the *BCL2L1* promoter identified via the UCSC Genome browser (https://genome.ucsc.edu/ accessed on 15 February 2021) that we cross-checked using the Lasagna tool browser (https://biogrid-lasagna.engr.uconn.edu/lasagna_search/index.php accessed on 15 February 2021). Among these, only few transcription factors have been confirmed to be able to bind Bcl-xL promoter and/or to regulate Bcl-xL expression. The first transcription factor identified in that respect was ETS2, belonging to the family of ETS. It was revealed by a specific co-expression of ETS2 and Bcl-xL in the same population of developing T-cells [61]. ETS2/Bcl-xL co-expression has been also identified in CD4+/CD8+ T cells, in primary bone marrow macrophages (BMM) derived from bone marrow progenitor cells, and in BMM upon activation of functional competence signals [62]. When expressed in human embryonic kidney cells or expressed in macrophages, ETS2 was able to upregulate Bcl-xL protein expression, thus rending these cells resistant to apoptosis [61].

NF-kB was found to upregulate *BCL2L1* expression, influencing its expression as evidenced for CD40 survival signals in human B lymphoma cells [63]. Moreover, c-Rel, a subunit of the NF-kB family, has been reported to be responsible for Bcl-xL expression. These results demonstrate that some death antagonists of the Bcl-2 family can affect oncogenesis through the effect of NF-kB on their expression [64].

The STAT family is activated after phosphorylation by JAKs and acts as a downstream effector of different cytokines or growth factors [65]. Different results have reported the involvement of several members of the STAT family in the transcriptional regulation of the *BCL2L1* gene, both in normal cells, such as hematopoietic and myoblast cells, and cancer cells. The first involvement of STAT3 in the regulation of Bcl-xL has been proved by Grandis and colleagues throughout the identification of a correlation between the constitutive activation of STAT3 and a higher expression of Bcl-xL in squamous cell carcinomas of the head and neck [66]. On the other hand, a reduced expression of STAT3 caused a decreased *BCL2L1* expression and induced apoptosis, while a functional STAT3 was necessary for *BCL2L1* reporter gene activity in myeloma cells, suggesting that STAT3 activity underlies the high expression of Bcl-xL detected in these tumors [67]. A positive correlation between *BCL2L1* expression and phosphorylated STAT3 was also identified in melanoma models [68]. In physiological conditions, the regulation of *BCL2L1* by STAT5 has been proved in hematopoietic cells [69]. Studies showing that STAT1, but not STAT3, mediated the expression of the *BCL2L1* gene upon treatment of cardiac myoblasts with leukemia inhibitory factor were also reported [70].

c-Fos and c-Jun are also involved in the regulation of *BCL2L1*. Jun proteins can homo-dimerize, whilst Fos proteins can only hetero-dimerize. They can form a complex defined as AP1 and regulate the transcription of target genes, whose promoter contains the AP1 binding sites. Further, the AP1 complex was found to affect both proliferation and differentiation pathways [71]. AP1 activity has been dubbed potentially crucial in controlling anti-apoptotic genes [72]. Presence of an AP1 site upstream of the first non-coding exon has been detected on the *BCL2L1* promoter, and c-Fos and c-Jun have been shown to be able to trans-activate the *BCL2L1* gene [61].

The splicing control of *BCL2L1* is central to regulate apoptotic response in normal development. As previously stated, *BCL2L1* gene can be alternatively spliced to produce the long-form Bcl-xL and the short-form Bcl-xS. Because of their different roles, the regulatory mechanisms of *BCL2L1* splicing have been largely investigated. A study conducted by Moore, MJ et al. identified several components of the spliceosome complex that are directly involved in *BCL2L1* splicing [73]. The study highlighted the fundamental role played by the alternative splicing network to regulate cell cycle control under normal and mitotic stress [73]. Specifically, the authors identified Small Nuclear Ribonucleoprotein Polypeptide N (snRNP) and splicing factors including U1 snRNP protein (SNRP70); U2 snRNP proteins (SF3B1, SF3B4, SF3B5, SF3A1, SF3A3, SNRNPA1); U5 snRNP proteins (U5-200K, PRPF6, UPS39); spliceosomal (Sm) and like Sm (LSM) core proteins (SmB/B′, SmD1, SmD2, SmD3); survival motor neuron (SMN) complex proteins (SMN1, GEMIN4: Gem-associated protein 4) [74].

Studies conducted in the last decade have identified also other RNA binding proteins that enhance or inhibit *BCL2L1* splicing, with consequent effect on apoptosis. These factors have been classified in 4 families [74]:i.Heterogeneous nuclear ribonucleoproteins (hnRNP) that are a large family of RNA binding proteins (RBPs) that control multiple processes in RNA metabolism and whose function depends on their ability to bind the RNA and their localization. Six different hnRNP regulating *BCL2L1* splicing have been identified. Among them, four have been reported to promote the production of Bcl-xS and two the production of Bcl-xL [75].ii.The serine/arginine-rich (SR) protein family that is a group of RBPs that contains a characteristic C-terminal arginine and serine-rich domain (RS domain) and one or two N-terminal RNA recognition motifs (RRMs) [76]. Different SR proteins have been identified as a key regulator in the splicing of *BCL2L1*. The most important of these proteins is the SRF10, which is involved in DNA damage, apoptosis, DNA repair and it has been shown to promote Bcl-xS production [77].iii.Signal transduction and activation of RNA (STAR) proteins, which constitute a family with a conserved K homology-domain and a STAR domain, responsible for the RNA recognition [78,79]. The STAR proteins can regulate the metabolism of the RNA, including the splicing. To date, the only STAR protein involved in the regulation of *BCL2L1* splicing is SAM68, that when overexpressed is able to increment the isoform Bcl-xS, whilst when depleted induces an accumulation of Bcl-xL [80].iv.The RNA binding proteins, which possess the RRMs [81]. Within this group of proteins, the RBM25 has been shown to specifically interact with a sequence in exon 2 of *BCL2L1* promoting the pro-apoptotic Bcl-xS 5′ splice site selection [82], while RBM11 plays an important role in the regulation of alternative splicing for neuron and germ cell differentiation [81]. RBM4 performs important functions in the inhibition of tumor progression. Specifically, it induces cancer cell apoptosis by modulating *BCL2L1* splicing and shifting to the pro-apoptotic Bcl-xS isoform. Additionally, RBM4 can also antagonize the oncogenic SR protein SRSF1 to regulate *BCL2L1* splicing and inhibit cancer cell growth [83].

Finally, in the last decades, different microRNAs (miRNAs) were found to target and decrease the expression of the anti-apoptotic Bcl-2 family members [74]. To date, most miRNAs identified are able to target only one of the anti-apoptotic family members even if the list of miRNAs capable of targeting different members increases over time. At least we can count three different miRNAs able to target different members of the Bcl-2 family: the miR-125b, whose targets are Bcl-2, Mcl-1 and Bcl-w [84,85], miR-133a/b able to target Mcl-1 and Bcl-xL [86], and miR-153 that targets Bcl-2 and Mcl-1 [87]. Among the miRNAs able to act against Bcl-xL we cite let-7c/g, which negatively regulates Bcl-xL expression [88]; miR-491 which is able to induce apoptosis by targeting Bcl-xL in different tumor types such as glioblastoma, ovarian and colorectal cancer [89,90,91]; miR-34a that is able to target Bcl-xL [92]; and miR377 that represses Bcl-xL [93].

Other modifications that Bcl-xL protein can undergo are phosphorylation mediated by SAPK/JNK causing an inhibition of *BCL2L1* expression [94,95], as it has been demonstrated for Bcl-2 and Mcl-1 [96].

### 3.3. Bcl-xL Expression and Canonical Functions in Melanoma

As previously discussed, the apoptotic process is unquestionably a necessary tool for tissue regeneration and long-lasting health, but at the same time, its alteration has been a strategy widely exploited by cancer cells as a defense mechanism against multiple therapies. In fact, tumor cells are able to escape apoptosis, ensuring an adaptation to the microenvironment and therapies. Melanoma cells feature a distinctive labile and stage-dependent phenotype, which is why pro-survival molecules can protect them from apoptosis and mediate other processes, thus increasing an aggressive phenotype.

By analysis of samples from human benign nevi, primary melanoma, and melanoma metastases in comparison with normal skin, the study of Ulrike Leiter and colleagues demonstrated that Bcl-xL was expressed in all metastatic melanoma samples, 80% of nevi and 62% of normal tissue samples, illustrating how Bcl-xL increases its expression passing from primary to metastatic melanoma [97]. The involvement of Bcl-xL in melanoma progression was also proved by Liqing Zhuang’s group via immunohistology analysis of Bcl-xL and other anti-apoptotic proteins, as Bcl-2 and Mcl-1, in sections of benign nevi, primary melanoma and metastatic melanoma [68]. In particular, even if Bcl-xL was expressed in benign nevi and thin melanoma, its expression was higher in sub-cutaneous and lymph node metastases when compared to benign nevi and thin melanoma. A positive correlation between Bcl-xL and tumor thickness or mitotic rate was also evidenced. An enhanced expression of Bcl-xL and Bcl-2 proteins passing from primary to metastatic melanoma has also been shown by Zhang H. and colleagues [98]. By using primary cell cultures derived from melanoma specimens, established cell lines and normal melanocytes from healthy donors, high Bcl-xL expression was observed in all melanoma samples tested by the group of Olie R.A. [99]. Interestingly, the expression of the pro-apoptotic Bcl-xS isoform has been reported to decrease during melanoma progression [97]. From all these findings, it is evident that elevated expression of Bcl-xL is associated with melanoma progression from primary into metastatic melanoma.

The canonical role of Bcl-xL as an anti-apoptotic factor in melanoma was fully elucidated in a recent study by Erinna F. Lee et al. [100], with the aim of evaluating the role that the different components of the Bcl-2 family play in melanoma. The authors conducted a study testing different BH3 mimetic drugs designed to target individuals or sub-groups of pro-survival Bcl-2 proteins, alone and in combination, in both 2D and 3D cell cultures in a panel of established and early-transition patient-derived cell lines. The study demonstrated that none of the drugs showed significant effects on a stand-alone basis, while combinations of drugs targeting Mcl-1 and Bcl-xL had a synergistic ability to kill cancer cells, hence providing evidence of how Bcl-xL and Mcl-1 appear to be key factors in maintaining melanoma cell survival. This study also shows a clear dissociation between changes in Bcl-2 expression (downregulation) and Bcl-xL or Mcl-1 expression (upregulation) during progression of melanoma. A different involvement of Bcl-xL and Bcl-2 was also reported in the study of Zhang and Rosdahl, indicating that Bcl-xL, but not Bcl-2, was a key protein in the induction of apoptosis after ultraviolet-B exposure in melanoma cells [98]. On the contrary, the study of Olie and collaborators performed using antisense oligonucleotides directed against either Bcl-xL mRNA or the Bcl-2 and the Bcl-xL mRNAs simultaneously [99] demonstrated that both the Bcl-xL monospecific oligonucleotide and the Bcl-2/Bcl-xL bispecific oligonucleotide reduced tumor cell viability by induction of apoptosis, but the bispecific oligonucleotide proved to be superior to the monospecific ones.

Although there are no clear explanations of how Bcl-xL could confer chemo-resistance, the aggressiveness induced by an over-expression of Bcl-xL has been often associated to its capacity to induce drug resistance in cancer from different origins [100,101,102]. Regarding melanoma, it has been demonstrated that forced expression of ectopic Bcl-xL converted drug-sensitive cell lines into drug-resistant ones [103]. Bcl-xL contributes to melanoma chemo-resistance through the protection from drug-induced apoptosis [103,104]. Reduction of Bcl-xL protein expression by specific antisense oligonucleotide enhanced the chemo-sensitivity of melanoma cells as well as chemotherapy-induced apoptosis. These data suggest that Bcl-xL is an important factor contributing to the chemo-resistance of human melanoma and can be inhibited by antisense therapy. Bcl-xL’s ability to induce chemo-resistance has also been attributed to the interaction between Bcl-xL and the Insulin Growth Factor (IGF1) [105]. Further, the specific silencing of both Bcl-xL and IGF1 by small interfering RNA evidenced the protective effect of IGF1 and a correlation with STAT5, suggesting promotion of anti-apoptotic chemo-resistance mechanism via the activation of the anti-apoptotic protein Bcl-xL [105]. Furthermore, the overexpression of Bcl-xL also blocked the cytochrome C release induced by anticancer drug, promoting cell drugs resistance and survival [106]. Finally, a downregulation of Bcl-xL after ultraviolet B radiation has been reported in primary melanoma samples, whereas matched metastatic specimens expressing higher Bcl-xL level were not affected by the treatment [98].

### 3.4. Non-Canonical Functions of Bcl-xL in Melanoma

In addition to its well-known role in canonical pathways such as apoptosis, and survival in cancer, Bcl-xL also affects other pathways, independent from survival and apoptosis, such as invasion/migration, epithelial mesenchymal transition, metastasization, and stemness (Figure 4)**.** These biological functions have been observed in tumors from different origin, such as glioma, breast, colorectal and pancreatic cancer [107,108,109,110]. In some cases, the role of Bcl-xL in the nucleus, but not in the mitochondria, has been identified as responsible for its non-canonical functions [107]. Moreover, the non-canonical functions of Bcl-xL have been reported by our group in melanoma. Recently, we have demonstrated that Bcl-xL incremented the in vitro cell migration and invasion in melanoma models, facilitating at the same time the formation of a vasculogenic structure [111]. Analysis of Bcl-xL overexpressing cells proved their ability to form tumor spheres associated to a pattern of stemness markers, supporting the idea that Bcl-xL is also important for the maintenance of cancer stem cell phenotype. In the past years, we focused our attention on the role of Bcl-xL in the regulation of angiogenesis in melanoma models. In particular, we found that conditioned medium of Bcl-xL overexpressing cells increased in vitro endothelial cell functions and in vivo vessel formation through the pro-angiogenic factor interleukin-8 (CXCL8). The use of neutralizing antibodies against CXCL8 confirmed the role of this chemokine in the angiogenesis induced by Bcl-xL overexpression. Downregulating Bcl-xL through antisense oligonucleotide or siRNA, confirmed the involvement of Bcl-xL in the expression of CXCL8 both at the protein and mRNA level. When melanoma cells were grown under low oxygen condition (hypoxia), we also highlighted Bcl-xL ability to increase the expression of the hypoxia inducible factor (HIF-1) and its target genes, vascular endothelial growth factor (VEGF) and metalloproteases 2 [112]. In fact, although in normoxic condition Bcl-xL did not affect the HIF-1/VEGF pathway, under hypoxia Bcl-xL, overexpression resulted in a higher level of HIF-1α and VEGF [111]. These data are consistent with the ability of Bcl-2 protein to cooperate with hypoxia to induce angiogenesis through VEGF in cancer models [111,113]. All these results corroborated the previous reported findings obtained in vitro and in vivo showing a unidirectional crosstalk able to empower the angiogenic phenotype of the endothelial cells [114]. In particular, in endothelial cells Bcl-xL was able to promote the release of VEGF, whose bond with its receptor VEGFR induced the expression of Bcl-2 and consequent induction of interleukin-1 and CXCL8, and activation of the autocrine signaling pathways enhancing the angiogenic activity.

To further investigate the involvement of Bcl-xL in angiogenesis, we also analyzed whether the Bcl-xL/CXCL8 pathway was important to promote angiogenesis and aggressiveness in zebrafish melanoma models [115]. Bcl-xL overexpressing melanoma cells showed enhanced dissemination and higher angiogenic activity in zebrafish embryos. Human CXCL8 protein was also able to induce a strong pro-angiogenic activity in zebrafish embryos, and using a morpholino-mediated gene knockdown, the CXCR2 receptor was identified as the mediator of CXCL8 pro-angiogenic activity. What stood out from our results was how aggressiveness of melanoma cells overexpressing Bcl-xL was mediated by an autocrine effect of CXCL8 on its receptor. Finally, via microarray and RNA seq public databases, a correlation between CXCL8 and markers of melanoma aggressiveness was also identified, together with a correlation between Bcl-xL and CXCL8 expression and poor prognosis of melanoma patients [115].

Lastly, Bcl-xL results involved also in autophagy, a process by which damaged or old cells undergo degradation [116]. Just as apoptosis, autophagy is essential to maintain tissue homeostasis but, at the same time, a deregulated process is often observed in pathological conditions, including tumors [117,118]. Although apoptosis and autophagy occur for different reasons, they share common stimuli as well as regulatory proteins, including p53, beclin1 and the Bcl-2 family. Several studies have demonstrated the ability of Bcl-xL to bind beclin1, a key regulatory factor indispensable for the formation of the autophagic complex [119]. In particular, the binding of these two proteins determines an inhibition of autophagy, thanks to the presence of a BH3 domain on beclin1, which can bind the hydrophobic groove of Bcl-xL [120]. It is particularly interesting how Bcl-xL under normal conditions protects cells from autophagy by inhibiting the Beclin-1-Vps34 complex, while in cancer cells induces autophagy by interacting with mitochondrial ARF tumor suppressor. As suggested by the authors, a possible explanation is that when the expression of ARF protein increases, Bcl-xL, binds ARF, thus leaving free Beclin-1-Vps34 complex to act [121].

## 4. Preclinical Studies of Bcl-xL Specific Inhibitors on Melanoma

Because of its relevance in the progression of cancer from different histotypes, different strategies have been evaluated in order to inhibit Bcl-xL. These include antisense oligonucleotides, small molecules including BH3 mimetics, Proteolysis Targeting Chimeric (PROTAC) molecules.

The first approach used to inhibit Bcl-xL was the development of antisense oligonucleotide targeting the mRNA of Bcl-xL or the mRNA of different antiapoptotic proteins. We previously demonstrated that antisense oligonucleotides against both Bcl-xL and Bcl-2 were able to inhibit angiogenesis in melanoma cell lines overexpressing Bcl-2 [122]. The same effect was also confirmed by Olie and colleagues, who used antisense oligonucleotides targeting either the Bcl-xL mRNA or the Bcl-2 and the Bcl-xL mRNA on primary cell cultures, stable cell lines and normal melanocytes from healthy donors [101,123]. They proved that both Bcl-xL monospecific oligonucleotide and Bcl-xL/Bcl-2 bispecific oligonucleotide effectively reduced tumor cell viability by induction of apoptosis. Bcl-xL/Bcl-2 bispecific antisense oligonucleotide also triggered p53-independent apoptosis in human melanoma cells [124]. Interestingly, the strategy with antisense oligonucleotide was successfully applied also in vitro and in vivo in combination with cisplatin in tumor models from different origin not including melanoma, such as mesothelioma [125], transitional cell carcinoma [126] and bladder carcinoma [127].

Another strategy used to inhibit Bcl-xL and other anti-apoptotic proteins is represented by small molecules able to abrogate the expression of anti-apoptotic members, mimicking the action of the BH3-only proteins. Initially, the formulation of these BH3 mimetic was against all the Bcl-2 anti-apoptotic members. One of the first small molecules used in melanoma was the pan BH3 inhibitor obatoclax (GX15-070), showing affinity for Bcl-2, Bcl-xL, Mcl-1, Bcl-w and A1/Bfl-1 [128]. Another small molecule inhibiting Bcl-2, Bcl-xL and Bcl-w is represented by ABT-737, whose effect on melanoma was demonstrated because of its ability to empower the efficacy of several therapeutic strategies including immunotoxins [129] and BRAF or MEK inhibitor in BRAF-mutated cells [130]. As it is not orally available, ABT-737 was replaced by ABT-263 (Navitoclax). The efficacy of Navitoclax has been reported for the in vivo treatment of BRAFV600E melanoma models in combination with copper chelators, able to sequester copper required for MEK1 and MEK2 activity through a direct copper–MEK1/2 interaction [131]. Unfortunately, the use of Navitoclax was restricted because of the side effect that it caused [132]. Nevertheless, a phase I/II trial study using Navitoclax in combination with dabrafenib (BRAF inhibitor) and trametinib (MEK inhibitor) is recruiting unresectable or metastatic patients with BRAF mutant melanoma (ClinicalTrials.gov, NCT01989585).

The first BH3 mimetic small molecule described to act specifically on Bcl-xL has been WEHI-539 [133]. It was identified by structure-guided design of selective Bcl-xL inhibitors and showed more than 400-fold selectivity for Bcl-xL when compared to other Bcl-2 antiapoptotic family members [134]. WEHI-539 induced a synergistic effect in melanoma models when combined with mitochondrial chaperone, G-TPP [135]. Its use was limited because of the poor physicochemical properties and high in vivo toxicity [133,136].

In recent times, the formulation of the first-in-class Bcl-xL inhibitor, A-1331852, has been described. It was produced by re-engineering BH3 mimetic Bcl-xL inhibitor, A-1155463, using structure-based drug design. A-1331852 is an orally active compound able to induce apoptosis in Bcl-xL-dependent tumor cells [137]. Unfortunately, there are still no preclinical tests able to provide efficiency of A-1331852 treatment in melanoma. Regarding A-1155463, while it did not affect melanoma cell viability, it increased the effect of immunotoxin targeting the melanoma-associated chondroitin sulfate proteoglycan 4 (CSPG4) when combined with the Mcl-1 specific inhibitor, S63845 [129].

Recently, DT2216, a Bcl-xL PROteolysis-TArgeting Chimera (PROTAC) derived from ABT-263, has been reported to be a potent inhibitor of different forms of cancer and with low toxicity versus platelets. The effect of DT2216 was also demonstrated in vivo where the outcome was proved both as a single agent and in combination with chemotherapeutic drugs [138]. No data are available regarding the effect of this chimera in melanoma models.

## 5. Conclusions

Melanoma represents the deadliest form of skin cancer. Features of melanoma are the high aggressiveness and the poor responsiveness to standard therapeutics.

Despite the great advances that have been made in the last years thanks to the advent of both targeted therapies, designed against factors of the BRAF/MEK pathways, and immunotherapy-blocking immune checkpoints, significant safety issues remain in relation to the activation of a toxic response in the immune system of the host and in the emergence of drug resistance. Several clinical trials are ongoing to overcome these limitations and to evaluate in unresectable or advanced melanoma the effect of immune checkpoint inhibitors in combinatorial regimes with targeted therapy (NCT01767454), other immune modulators such as interleukin-2 (ClinicalTrials.gov Identifier: NCT0456212) and IDO (ClinicalTrials.gov Identifier: NCT02073123), therapeutic cancer vaccine (ClinicalTrials.gov Identifiers: NCT04382664, NCT04697576) or stereotactic body radiation therapy (ClinicalTrials.gov Identifier: NCT03693014).

Molecular mechanisms leading to melanoma development, progression and response to therapy are the focus of intense investigation, aimed at understanding its pathobiology, the molecular mechanism of drug resistance and, finally, at developing new treatment strategies. Understanding the factors that promote progression of melanoma could help physicians to predict its probable course, thereupon planning treatment and anticipating further care.

Among Bcl-2 family proteins, Bcl-xL has emerged as a multifaceted factor that plays a pivotal role in the progression of different kinds of cancer, including melanoma, acting at different stages as initiation, invasion, metastasis, and acquisition of chemo-resistance. As Bcl-xL, and other anti-apoptotic proteins, could represent possible targets for melanoma therapy alone or in combination, the search for specific agents able to target these proteins is still ongoing. Trial studies are ongoing to investigate the effect of bcl-2 family pan inhibitors in combination with BRAF and MEK inhibitors in melanoma patients (ClinicalTrials.gov Identifier: NCT01989585).

In conclusion, together with the role that single components of the Bcl-2 family play in tumor progression and response to therapy, the significance of their balance is left to be defined, together with a correlation among the single factors. In fact, many data show how it is the imbalance between the Bcl-2 homologues and their pro-death counterparts to constitute an advantage for tumor cell. Further studies are also necessary to understand the role of single or multiple Bcl-2 family components expressed by tumor cells within the tumor microenvironment.

## Figures and Tables

**Figure 1 ijms-22-02777-f001:**
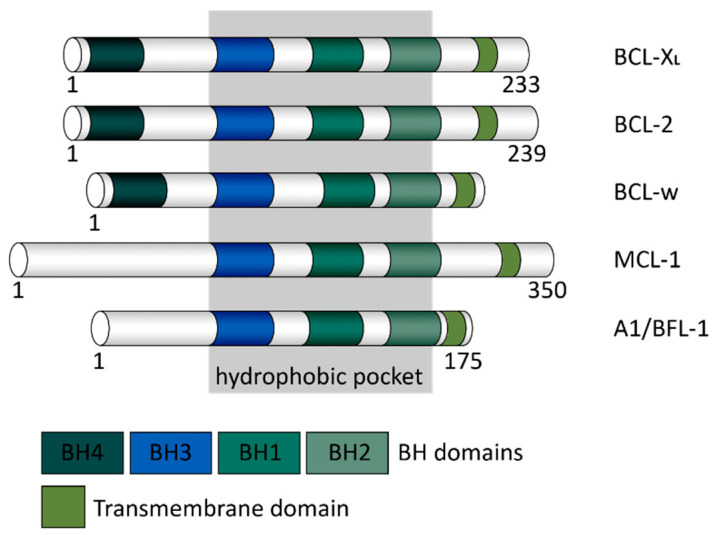
Schematic representation of the anti-apoptotic Bcl-2 family members. Bcl-2 homology (BH) and transmembrane domains are represented in distinct colours. The hydrophobic pocket is marked in grey.

**Figure 2 ijms-22-02777-f002:**
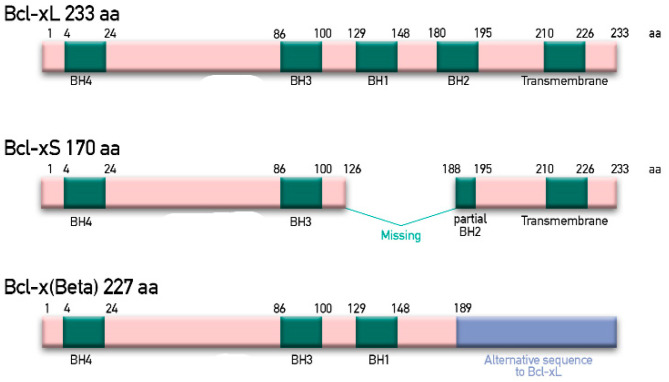
Schematic representation of the alternative spliced variants Bcl-xL, Bcl-xS and Bcl-xβ. Bcl-2 homology (BH) and transmembrane domains are represented in green. Alternative sequence present in Bcl-x(Beta)(Bcl-xβ) is reported in violet. No defined domains are shown in pink.

**Figure 3 ijms-22-02777-f003:**
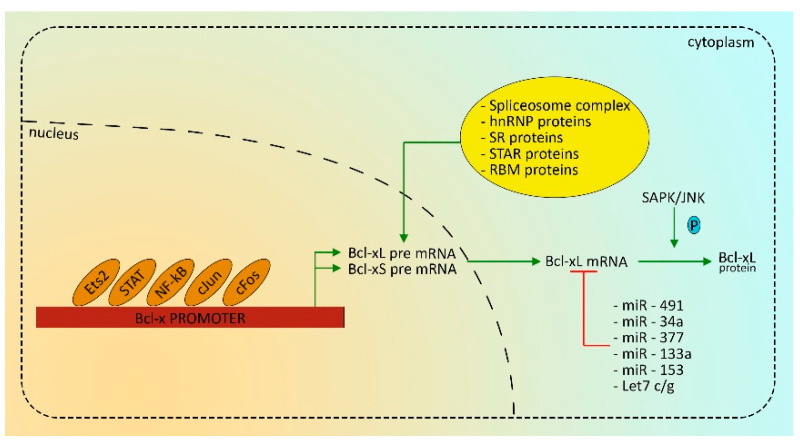
Transcriptional factors and effectors involved in the Bcl-xL regulation.

**Figure 4 ijms-22-02777-f004:**
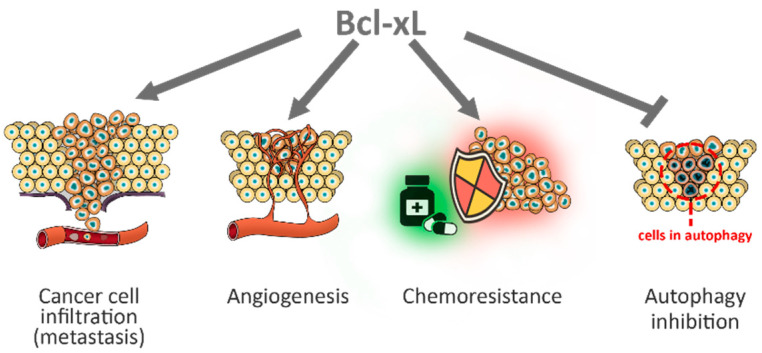
Non-canonical role played by Bcl-xL on melanoma. Bcl-xL is involved in metastasis, angiogenesis, chemoresistance against different chemotherapeutic drugs, and autophagy.

## Data Availability

Not applicable.

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
