# Peer review of "Bcl-xL: A Focus on Melanoma Pathobiology"

_ijms, 2021, doi:10.3390/ijms22052777_

Round 1

Reviewer 1 Report

Malignant melanoma is one of the most difficult cancers to treat due to its resistance to chemotherapy. In recent decades, emerging molecular-targeted therapies and immune checkpoint inhibitors for patients with metastatic melanoma. Immune checkpoint blockade has revolutionized the treatment of patients with advanced melanoma and many other cancers. Blockade of inhibi­tory receptors, CTLA-4 and PD-1, enhances T-cell-mediated antitumor immune responses, leading to improved survival and durable responses in patients.

In this review, the authors do not describe and discuss the new approaches and combination therapies that are being developed to broaden the clinical impact of immune checkpoint blockade by overcoming resistance to therapy and limiting adverse events; while the authors pay particular attention to the members of the Bcl-2 family (paragraph 4. Bcl-2 famil: is very long and verbose, especially the regulation of Bcl-xl). Also, the in vivo efficacy of target-specific combinations should be discussed by reporting the numerous promising results especially in discussion and conclusions.

Author Response

As suggested by the reviewer we i) described and discussed the new approaches and combination therapies that are being developed; ii) shortened “paragraph 4. Bcl-2 family”; iii) included in introduction and conclusion the efficacy of target-specific combinations.

Reviewer 2 Report

In their publication, authors discussed extensively the role of the BCL-xL protein in human melanoma pathobiology. First, authors provided background information on the structure, function and regulation of the Bcl-xL protein, then focused on its contribution to the development and aggressiveness of melanoma. Very interesting was the discussion of its non-canonical functions in melanoma, including cell migration and invasion. Finally, authors provided some information about the current research on the possibility of targeted cancer therapies based on inhibition of Bcl-xL.

Broad comments: 

Generally, the paper is well composed and allows the reader to slowly familiarize with the subject. It is easy to read in some parts, however, in general, the paper appears to be underdone. There are no tables and figures quoted by the authors or they are incorrectly numbered. Bibliographic data in the reference list needs a thorough review. In many places paper requires a profound stylistic and English language correction. There are also a few substantive errors. However, they are not major and can be easily corrected. The abstract, apart from minor inaccuracies listed below, is very well written and reflects the content of the article.

Specific comments :

Line 14 – Only by alternative splicing? You mentioned in the text also other mechanisms of Bcl-xL regulation. Maybe just add “i.a.”.

Line 17 - You didn't discuss the role of Bcl-xL in epithelial to mesenchymal transition in the text, why are you listing it here? Instead, you do not mention its role in angiogenesis, what you broadly describe in the text.

Line 30 - To avoid repeating the word "constant" in the next two sentences, I would suggest removing the word before "homeostasis".

Line 35-36 - Could you change this sentence? It doesn't sound good two sentences that end with "cancer".

Line 42-43 – Ignited inactive caspases activate effector caspases? I know what you mean, but it sounds confusing. Please change this sentence to make it clear that caspases are activated in a cascade process that ends up in activation of the effector caspases.

Line 49 – Only one ligand?

Line 50 – In my opinion the sentence would benefit if you remove "production of ligand" and leave only “After death ligand binding to the death receptors expressed in the target cell membrane”. It was already mentioned where do the ligands come from.

Line 59 – It does not contain, it forms apoptosome by oligomerization

Line 65 – Can you change word “represent” to “account for”. First, in my opinion it sounds weird that cancer represent death. Second, you use the same word in the next sentence.

Line 66-69 – Lack of the source of this information. Can you provide the reference?

Line 76 - Lack of the source of this information. Can you provide the reference?

Line 77 – 83 - Lack of the source of this information. Can you provide the reference? If it is reference 26, move it to the end of paragraph.

Line 84-86 – Again, lack of the source of this information.

Line 88 – In my opinion “difficult” would be the better word than “hard”. “…,thus causing death” – Could you express it differently? It doesn't sound good in this form.

Line 94 – “often associated with the use of chemotherapy”. Could you also express it differently?

Line 95 – “treatment could include side effects due to….” - also does not seem to be a correct expression. Please, change it.

Line 95 – “the toxicity at the level of…” I have not met such an expression before. At least in that context. In my opinion, level of toxicity means entirely something else.

Line 96-97 - In my opinion, the emergence of drug resistance cannot formally be included in the side effects of chemotherapy, especially since the source you refer to says nothing about it. Please state it differently and provide appropriate reference.

Line 101 – What mechanisms are you talking about? It is incomprehensible.

Line 102- 142 - In my opinion, paragraph 3 about Zebrafish as a model is completely unnecessary. It is not related to the topic described in the article, it is only a research method that allowed to acquire the knowledge described in the article. If you wish to leave it, please limit it to the general information about the role of this model in melanoma studies, without going into the details regarding, for example, interaction between tumor cells and tumor-associated macrophages, MITF and the like, completely unrelated to the topic.

Line 116-117 - Relationship between transparency of zebrafish embryos and possibility to observe particularly the influence of the environment on melanoma cells and vice versa is not clear. In my opinion, this allows to observe the development of melanoma under the influence of many other factors, not only the environment and much more. And that's how it is worded in the next sentences concerning adult zebrafish lacking body pigment cells. Could you rephrase this information more generally, saying that this model, due to its transparency, allows for the in vivo observation of melanoma development, its biology and characteristics?

Line 152 – “present in these proteins”- it has already been said that this domain is present in these proteins, it makes no sense to repeat it in the next sentence. Could you please delete this?

Line 152-158 - It is not clear why you describe these details about this domain in this moment. This is a bit inconsistent. If you want to describe these details about BH4 domain here, give a few words of introduction about the importance of this domain for a reader less familiar with the topic. In my opinion, it would be beneficial if you move the information from line 195-199 before line 152 and combine information about this domain in one place.

Especially, information from line 152-157 – It is not clear if it is the only role of this domain or only additional. It is confusing in this moment for the reader.

Line 157-158 – This information confirms that this domain is responsible for the interaction of pro-apoptotic proteins with the BCL-xL and is a supplement to the information contained in the lines 195-199. Introduced before it is confusing.

Line 164 - The information that BAK and BAX proteins are important is obvious and doesn't sound good.

Line 168-170 – Could you change this sentence. It doesn't sound good.

Line 172 – Could you provide the examples of these synthesizers? And could you briefly mention their role? I know it is explained in the last sentence, but not directly, and you gave a note that they do not bind directly to BAX and BAK, so it raises the question for the reader what exactly they do.

Line 187 – “Bcl-xL protein level structure” – Is it correct expression?

Line 200 – 204 – Could it be explained in further details? You say it is interesting and distinguishes the individual pro-survival protein, but was it found if this fact have any significance?

Line 221-229 – This sentence is incomprehensible. Shouldn’t it be “among which” instead of “in which”?

Line 225 – Figure 3 is not related to transcription factors

Line 227 - Replace the number 2 with the word two

Line 231 – Table 1 is missing

Line 221-224 and line 230 – 236 – In my opinion this is the same information given two times. In the first paragraph (line 221-229) you mentioned that “BCL2L1 promoter contains consensus motifs for a number of transcription factors…”, and some of them (STATs, Rel/NF-kB, ETS and the AP1 complex) seems to be important in the regulation of Bcl-2-L1 gene expression. And again (line 230-236) you give the information that “By comparing mouse and human BCL2L1 promoter sequences, different consensus region binding sites for different transcriptional families were identified” and “Among these, only few transcription factors have been confirmed to be able to bind Bcl-xL promoter and/or to regulate Bcl-xL expression”. I would suggest removing lines 221-224 or rewriting this paragraph.

Line 243-245 – I don't understand this information. What does it mean that this is “the most plausible mechanism”? How else could a transcription factor upregulate expression if not by activating transcription?

Line 246-250 - Could you please rewrite the paragraph on the role of the NF-kB in the regulation of Bcl-xL transcription? Especially the first two sentences in their present form seem to be separate from the rest of the text. In my opinion, it would sound better if you just write that NF-kB can also influence Bcl-xL expression as evidenced by the two cited studies.

Line 258-263 – You describe various methods that researchers used in this publication to prove the role of STAT3 in the regulation of Bcl-xL expression, which is unnecessary in my opinion. Better to describe the conclusions of this study mentioning how it was proven.

Line 263-264. Rephrase this sentence, please. It does not sound good. What does it mean in normal conditions?

Lines 268-269 – It sounds like extracellular stimuli belong to the basic/leucine zipper family, not c-Fos and c-Jun.

Line 277-278 – It is not clear how involvement in tumorigenesis, cancer progression and drug resistance explain role of splicing control in normal development.

Line 285-290 – It is not clear what this information is about. Identified the role of this factors in splicing of BCL2L1 or only generally identified them without link to BCL2L1 splicing?

Line 295, 332 – Could you explain the abbreviation RBPs and RRMs when they first appear in the text?

Line 304-305 – It is not clear if SRF10 and SFSF10 is the same protein or you've already discussing the next.

Line 299, 308, 391 - The appropriate reference should always appear at the end of the described studies not in the middle.

Line 327 – You have already discussed the other mechanism of regulation of expression of Bcl-xL such as transcription factors so it is obvious that alternative splicing is not the only mechanism. Did you mean post-transcriptional regulation of Bcl-xL?

Line 340 – It is not clear what causes an inhibition of BCL2L1 expression: Bcl-xL phosphorylation, SAPK/JNK or microtubule-damaging drugs and what was demonstrated for Bcl-2 and Mcl-1. Rewrite this sentence, please.

Line 366-368 – Could you please change this sentence? It sounds like the cells are a tool for observation, not its subject.

Line 375 – Shouldn’t it be Bcl-2 family or proteins?

Line 376 - Could you please check this sentence? It is the drugs that are tested against cells, not the other way around.

Line 386 – Could you specified this role of Bcl-xL in apoptosis induced by UV? In my opinion “response to apoptosis”  is not appropriate term, as Bcl-xL downregulation starts apoptosis pathway rather than is a respond to it.

Line 393 - The aggressiveness of Bcl-xL – it does not sound good. The protein cannot be aggressive in my opinion.

Line 399 – Please, put the appropriate reference at the end of the cited studies. It is not known where this information comes from.

Line 408 – What does it mean that Bcl-xL was sensitive to UVB? Did it cause its degradation? And how the information that metastatic specimens with higher Bcl-xL level were insensitive to UVB treatment prove this sensitivity?

Line 416 – There is no figure 4. It is figure 3 showing this non-canonical functions.

Line 425 – What is the sense of mentioning the year you focused on the role of Bcl-xL in the regulation of angiogenesis in melanoma models?

Line 437, 439 – Could you change one of the sentences to avoid repeating “these data” two times?

Line 447 – Models of what?

Line 457-458 – This sentence does not sound correct.

Line 467 – Correct Bclx-L to Bcl-xL.

Line 469-472 – Could you rewrite this sentence? Doesn't sound good. Besides, the reader already knows that Bcl-xL can bind to both proteins, there is no need to add it again.

Line 487 – The word “because” is unnecessary and confusing. This inhibitor was not described because of its ability to induce apoptosis in Bcl-xL overexpressing cells. It was first designed and described and then its ability to induce apoptosis in Bcl-xL overexpressing cells was confirmed.

Line 492 – In my opinion the word “antagonize” would be better than “abrogate the expression”.

Line 493 – “formulation of these compounds was against all the Bcl-2 anti-apoptotic members”. Please, change it. Doesn’t sound good.

Line 497 - Wouldn't the word "by" be more appropriate in this meaning than "because of"?

Line 509 – “Inhibitors of cancers” is not a commonly used phrase in science, although it appears to be correct. Remove “and”. It is unnecessary.

Line 510 – What outcome? The outcome can be positive or negative. Outcome as or outcome for?

Line 473-516 - This paragraph is chaotic. You begin this paragraph: different strategies have been evaluated in order to inhibit Bcl-xL, and you described the first one - antisense oligonucleotide, than you suddenly wrote about first inhibitor, not mentioning that this is the next strategy. And then you came to another strategy, which is the application of BH3-only mimetics to suddenly end with next inhibitor again. In this form, it is not clear to the reader what the difference between inhibitors and mimetics is and what you are actually talking about.

In my opinion, it would benefit if you combined information about inhibitors, described first their use in general as a strategy, describe the first (WEHI-539), then add information about the others if there are any. And you can end with this last one, explaining what differs it from the previous ones (this can be one word for example “highly selective”) or explain that it is the only one that was created besides WEHI-539. In this form, it is not very clear why you write about this particular inhibitor and not about others. And what does it mean it is “first class”?

Line 517 – It is “Conclusion” not “Discussion”.

Please check the references list again. There are a lot of mistakes in it. Standardize the way of providing bibliographic data in references.

Author Response

 As suggested by the reviewer we i) quoted and correctly numbered tables and figures, ii) modified the manuscript as requested in the Specific comments, iii) deleted paragraph 3 about Zebrafish, iv) reorganized paragraph 5 in a “less chaotic” ones; v) standardized the way of providing bibliographic data in references. Moreover, a native tongue person revised the manuscript.

Reviewer 3 Report

This is an interesting work in its field, with high quality explanatory images; however, some issues need to be addressed:

  • section 2, Melanoma: please refer to worldwide, rather than local incidence and prevalence rates
  • section 2, Melanoma: actually, a family history of melanoma is one of the main risk factors involved in its pathogenesis, due to the presence of high-penetrance melanoma susceptibility genes
  • section 2, Melanoma: in the radial growth phase, melanoma may invade the papillary dermis as well; please reformulate the sentence
  • line 118: "clinical features" instead of "features"
  • The authors should cite and discuss the timely review by Trisciuoglio and Bufalo doi: 10.1016/j.drudis.2021.01.027, and the recent work on ABT263 in uveal melanoma by Decaudin et al. doi: 10.1016/j.ejca.2019.12.012.
  • The Discussion section should be renamed to Conclusions
  • a thorough language revision is needed

Author Response

All the issues raised by the reviewer have been addressed, except for the inclusion of the recent work on ABT263 in uveal melanoma by Decaudin et al. doi: 10.1016/j.ejca.2019.12.012. We omitted this paper because the focus of the review was cutaneous melanoma.

Round 2

Reviewer 1 Report

This is an interesting work in its field. All issue have been addressed.

Reviewer 3 Report

All issues have been addressed.